# Lignin–Cobalt Nano-Enabled Poly(pseudo)rotaxane Supramolecular Hydrogel for Treating Chronic Wounds

**DOI:** 10.3390/pharmaceutics15061717

**Published:** 2023-06-13

**Authors:** Giulia Crivello, Giuliana Orlandini, Angela Gala Morena, Alessandro Torchio, Clara Mattu, Monica Boffito, Tzanko Tzanov, Gianluca Ciardelli

**Affiliations:** 1Department of Mechanical and Aerospace Engineering, Politecnico di Torino, Corso Duca degli Abruzzi 24, 10129 Torino, Italy; giulia.crivello@polito.it (G.C.); clara.mattu@polito.it (C.M.); monica.boffito@polito.it (M.B.); 2Group of Molecular and Industrial Biotechnology, Department of Chemical Engineering, Universitat Politècnica de Catalunya, 08222 Terrassa, Spaintzanko.tzanov@upc.edu (T.T.)

**Keywords:** supramolecular hydrogels, poly(pseudo)rotaxane, poly(ether urethane), α-cyclodextrin, cobalt-lignin nanoparticles, chronic wounds, inflammation, bacterial infection, matrix metalloproteases, myeloperoxidases

## Abstract

Chronic wounds (CWs) are a growing issue for the health care system. Their treatment requires a synergic approach to reduce both inflammation and the bacterial burden. In this work, a promising system for treating CWs was developed, comprising cobalt-lignin nanoparticles (NPs) embedded in a supramolecular (SM) hydrogel. First, NPs were obtained through cobalt reduction with phenolated lignin, and their antibacterial properties were tested against both Gram-negative and Gram-positive strains. The anti-inflammatory capacity of the NPs was proven through their ability to inhibit myeloperoxidase (MPO) and matrix metalloproteases (MMPs), which are enzymes involved in the inflammatory process and wound chronicity. Then, the NPs were loaded in an SM hydrogel based on a blend of α-cyclodextrin and custom-made poly(ether urethane)s. The nano-enabled hydrogel showed injectability, self-healing properties, and linear release of the loaded cargo. Moreover, the SM hydrogel’s characteristics were optimized to absorb proteins when in contact with liquid, suggesting its capacity to uptake harmful enzymes from the wound exudate. These results render the developed multifunctional SM material an interesting candidate for the management of CWs.

## 1. Introduction

Wound healing is a complex process requiring the synergic activity of cells, enzymes, and growth factors to restore skin integrity and functions. When the delicate balance governing this process is impaired, chronicity may develop. Chronic Wounds (CWs) are those that “have failed to proceed through an orderly and timely reparative process” [1] and can result in severe tissue injury and even death. The causes of this impairment are infection, ischemia, repeated trauma, and the presence of chronic pathologies such as diabetes, cardiovascular and cerebrovascular diseases, hypoxia, and cancer [2,3,4]. Over the last few decades, the progressive aging of the population in developed countries has resulted in a higher incidence of chronic pathologies with a consequent increase in cases of CWs. This phenomenon is a growing concern for the healthcare systems since it is estimated that the prevalence of CWs in the worldwide population is 2.21 per 1000 people [5], with an estimated cost in Europe of 2% of the national healthcare expenditure [6]. 

Despite having different causes, CWs are characterized by a permanent state of inflammation, presenting excessive levels of proinflammatory cytokines, reactive oxygen species (ROS), and proteases [7], and the frequent presence of infections, which worsen the inflammatory state and delay the healing process [8]. Excessive neutrophil activity plays an important role in sustaining CWs [9]. These cells release matrix metalloproteases (MMPs), which are a class of enzymes involved in the degradation of extracellular matrix (ECM) components such as collagen and elastin [9,10]. Moreover, neutrophils secrete myeloperoxidase (MPO), an enzyme involved in the production of hypochlorous acid (HOCl), which is known to degrade several biological molecules, including tissue inhibitors of MMPs (TIMMPs) [11,12]. MMP overexpression paired with TIMMP inhibition results in excessive ECM degradation, which further stimulates the inflammatory state, thus hindering the wound-healing process [7,9,10,13,14,15].

Given these premises, there is an increasing need for efficient CW treatments capable of simultaneously reducing inflammation by decreasing levels of MPO and MMPs and fighting the bacterial burden. Current treatments mostly rely on wound debridement and disinfection and the use of dressings containing antibacterial or analgesic compounds [16,17,18,19]. However, these approaches usually do not address the origin of inflammation. Moreover, an ideal wound dressing should be able to absorb excess exudate while maintaining a moist environment that promotes regeneration [20]. In the past few years, hydrogels have gained increasing attention due to their ability to absorb large volumes of biological fluids and their potential to be loaded with different therapeutic agents, such as analgesic and antibacterial compounds, that can be released in a controlled fashion [20,21,22,23]. Among the diverse possibilities, physically crosslinked hydrogels are particularly interesting since they are obtained through reversible interactions (e.g., Van der Waals, hydrophobic, or ionic bonds) and form under mild conditions, thereby minimizing the risk of drug degradation or denaturation [24]. There are different strategies that exploit specific interactions to induce the sol-to-gel transition, eventually leading to the formation of supramolecular (SM) structures. Among these strategies, the use of cyclodextrins has been broadly studied due to their ability to form host–guest complexes with different molecules [25]. In particular, α-cyclodextrins (αCDs) can interact with poly(ethylene oxide) (PEO) domains in linear polymers forming SM structures known as poly(pseudo)rotaxanes (PPRs) [26,27,28]. These PPRs have been reported to further organize into channel-like supramolecular crystals that trigger the sol-to-gel transition of bioartificial αCD/polymer aqueous solutions, resulting in stable gel networks [29,30]. Hydrogels based on PPR present thixotropic properties and can be used in a wide range of applications due to their high versatility and responsiveness [31]. However, their potential has yet to be fully explored in the field of smart drug delivery.

Recently, innovative hydrogels that exploit the above-described interactions were developed by our group [32,33] via the combination of αCDs with custom-made poly(ether urethane)s (PEUs). The PEUs were synthesized from Poloxamers^®^, which are amphiphilic triblock copolymers based on PEO and poly(propylene oxide) (PPO). The hydrogel’s gelation was driven by the combined effects of the PPR formation, the complexation of the αCDs with the PEO domains of the PEUs, and the organization of the PEUs amphiphilic chains into micelles. Using this approach, the polymeric content in the gel was greatly reduced from 10–15% *w*/*v* for the bare PEU thermosensitive hydrogels to 1% *w*/*v* for the αCD-PEU formulations, thereby minimizing potential cytotoxic effects while maintaining good mechanical properties and stability. The obtained hydrogels showed self-healing properties, injectability, and the capacity to store and control the release of encapsulated compounds [33]. In the present study, this hydrogel formulation was further optimized, and the obtained gel was used for the first time to encapsulate lignin–cobalt nanoparticles (CoLig NPs) for the treatment of CWs.

Lignin is an aromatic biopolymer obtained as a by-product of the paper industry. It is one of the most abundant biopolymers in the world, but its potential as a macromolecule is rarely valorized. Indeed, it can be used as a reducing agent for metals owing to its phenolic and aliphatic hydroxyl groups [34,35,36]. In this study, lignin was combined with tannic acid (TA) and L-tyrosine hydroxamate (LTH) through an enzymatic reaction mediated by the enzyme laccase. The addition of TA, a natural polyphenolic compound, was carried out to increase the number of phenols in the materials, which should improve their antioxidant activity and inhibitory action against MPO [36]. While, the hydroxamate groups grafted on lignin through LTH addition are known as MMP inhibitors [37,38]. 

Nanoparticles (NPs) were then obtained through the green reduction of cobalt via the phenolic groups of lignin and TA. This metal was chosen in order to increase the antimicrobial activity of phenolated lignin [39]. Metals are well-known antibacterial agents; silver is used as a gold-standard agent due to its high, broad-spectrum activity and its reduced propensity for facilitating bacterial resistance [40,41]. Nevertheless, allergies and hypersensitivity to silver necessitate the search for new alternatives for the development of medical treatments [42]. Cobalt is currently emerging as a promising agent for biomedical applications, showing activity against bacteria [43,44] and cancer cells [45]. Additionally, its magnetic properties render it a valid candidate for applications in sensors, imaging, and hyperthermia treatments [45].

The antibacterial activity, cytotoxicity, and ability to inhibit MPO and MMPs of cobalt–lignin NPs were validated. Then, the NPs were used as active fillers of the SM hydrogel, and the gel–NP interactions were studied through microscopy imaging, rheology, and stability tests in aqueous media at 37 °C. Finally, the release of the NPs from the hydrogel was investigated. The system showed injectability, self-healing properties, and protein uptake capacity along with improved mechanical properties and durability related to the gel–NP interactions. These results render the system an interesting candidate for the treatment of CWs. Moreover, the observed capacity of the NPs to stabilize the hydrogel network represents a promising feature to be further exploited in order to obtain physical hydrogels with enhanced mechanical properties.

## 2. Materials and Methods

### 2.1. Reagents, Materials and Cells

Protobind 6000 sulfur-free lignin powder was purchased from Green Value (Switzerland). Gallic acid, tannic acid, L-tyrosine hydroxamate, 3′,5′-dimethoxy-4′hydroxyacetophenone (97%) (acetosyringone), guaiacol, hydrogen peroxide (H_2_O_2_), 2,2′-azino-bis(3-ethylbenzothiazoline-6-sulfonic acid) diammonium salt (ABTS), phosphate-buffered saline (PBS), Folin–Ciocalteu phenol reagent, sodium carbonate, fluorescamine, dimethyl sulfoxide (DMSO), cobalt(II) nitrate hexahydrate, bovine serum albumin (BSA), nutrient broth (NB), Baird–Parker agar, cetrimide agar, and Dulbecco’s modified Eagle’s medium (DMEM) were purchased from Sigma-Aldrich (Madrid, Spain and Milan, Italy). Poloxamer^®^ 407 (P407, $\overset{-}{M_{n}}$ 12,600 Da) and Pluronic^®^ F68 (F68, $\overset{-}{M_{n}}$ 8400 Da); 1,6-hexamethylene diisocyanate (HDI) (168.2 Da); 1,4-cyclohexanedimethanol (CDM) (144.2 Da); N-Boc serinol (NBoc) (191.2 Da); dibutyltin dilaurate (catalyst, DBTDL); 3-(trimethylsilyl)propionic-2,2,3,3-d 4 acid sodium salt (TSP); and trifluoroacetic acid (TFA) were purchased from Merck (Milan, Italy). A-cyclodextrins (αCDs) were purchased from TCI Chemicals Europe (Zwijndrecht, Belgium). AlamarBlue cell viability reagent, myeloperoxidase, and EnzChek™ Gelatinase/Collagenase Assay Kit were purchased from Invitrogen, Life Technologies Corporation (Madrid, Spain). Novozym 51003 laccase from Miceliophtora termophila was provided by Novozymes (Bagsvaerd, Denmark). The enzymatic activity of laccase was 179 U/mL, which was defined as the amount of enzyme converting 1 μmol of ABTS to its cation radical (ε_436_ = 29 300 M^−1^ cm^−1^) in 5 mM Britton Robinson buffer (pH 5) at 40 °C. Bacterial strains (*Staphylococcus aureus* ATCC 25923 and *Pseudomonas aeruginosa* ATCC 10145) and human cell lines (fibroblasts ATCC-CRL-4001, BJ-5ta, and keratinocytes HaCaT cell line) were purchased from the American Type Culture Collection (ATCC LGC Standards, Spain). The water used in all experiments was purified using the Milli-Qplus system (Millipore) with 18.2 MΩ cm resistivity before its use. All solvents used for poly(ether urethane) synthesis, which were purchased from Carlo Erba Reagents (Milan, Italy), were of analytic grade and used as received, with the exception of 1,2-dichloroethane (DCE), which was dehydrated over activated molecular sieves overnight before PEU synthesis. All other reagents required for the synthesis of the polymers were dried according to the protocol reported by Boffito et al. [46].

### 2.2. Enzymatic Functionalization of Lignin 

Protobind 6000 lignin was functionalized with tannic acid (TA) and L-tyrosine hydroxamate (LTH) following a previously described protocol with some modifications [36]. Lignin (10 mg/mL) was soaked at 50 °C and stirred (250 rpm) in sodium acetate buffer (50 mM, pH 5), in which acetosyringone was previously dissolved at 1.5 mg/mL. Laccase (1% *v*/*v*) was incubated with the lignin solution for 1 h at 50 °C (with stirring applied) to initiate the oxidative reaction. Then, TA (10 mg/mL) and LTH (0.5 mg/mL) were added to the solution, and the reaction proceeded (with stirring applied) for 2 h at 50 °C. To separate the unreacted TA and LTH from the obtained material, the solution was centrifuged at 4750 rpm and 4 °C for 15 min. The obtained powder, referred to as Lig-TA-LTH, was freeze-dried and characterized. 

### 2.3. Characterization of Enzymatically Functionalized Lignin (Lig-TA-LTH)

The obtained Lig-TA-LTH was characterized to verify the successful conjugation of TA and LTH. Attenuated total reflectance–Fourier transform infrared (ATR-FTIR) spectroscopy was used to observe variations in the chemical structure of the material. Variations in the phenolic content were measured, using Folin–Ciocalteu reagent to confirm TA’s addition. LTH content was evaluated through fluorescamine fluorimetric assay. 

#### 2.3.1. Attenuated Total Reflectance–Fourier Transform Infrared (ATR-FTIR) Spectroscopy

To characterize the functionalized lignin, ATR-FTIR analysis was performed on Lig-TA-LTH using a PerkinElmer Spectrum 100 FTIR spectrometer (PerkinElmer, Waltham, MA, USA). Spectra were obtained as a result of an average of 32 scans taken within the spectral range from 4000 to 600 cm^−1^ at a resolution of 1 cm^−1^. Analyses were performed at room temperature. Lignin, TA, and LTH spectra were collected for comparison.

#### 2.3.2. Measurement of the Phenolic Content 

The phenolic content of Lig-TA-LTH was determined spectrophotometrically using Folin–Ciocalteu reagent. Briefly, 20 μL of a water solution containing Lig-TA-LTH (1 mg/mL) was mixed with 100 μL of 20% *w*/*v* sodium carbonate (Na_2_CO_3_) and 80 μL of 0.2 N Folin–Ciocalteu phenol reagent, and the resulting sample was incubated in the dark. After 10 min of incubation in the dark, the absorbance at 765 nm was measured using an Infinite M200 spectrophotometer (TECAN). Gallic acid (GA) was used to develop a calibration curve (0.1–1 mg/mL). All samples were measured in triplicate, and the results are shown in GA equivalents (GAE) per g of material.

#### 2.3.3. Quantification of the LTH Content

LTH presents a primary amine that was used to quantify the LTH content in the Lig-TA-LTH conjugates through fluorescamine fluorimetric assay. Briefly, 75 µL of the samples previously solubilized in water at 1 mg/mL was mixed with 25 µL of fluorescamine reagent dissolved at 3 mg/mL in DMSO. The plate was incubated at room temperature for 10 min; then, fluorescence at λ_ex/em_ = 390/470 nm was measured with an Infinite M200 spectrophotometer (TECAN). LTH was used to create a calibration curve. All samples were measured in triplicate.

### 2.4. Preparation of Cobalt/Lignin Nanoparticles (CoLig NPs)

CoLig NPs were obtained by modifying the protocol reported by Morena et al. [36]. Lig-TA-LTH was dissolved in water (5 mg/mL), and the pH was adjusted to 8.5. Then, 30 mL of the solution was mixed with 20 mL of cobalt(II) nitrate hexahydrate solution (4 mg/mL in water) and incubated for 2 h at 60 °C with stirring applied. The NPs were purified via centrifugation at 18,000 rpm for 20 min. The obtained pellet was resuspended in water and sonicated for 30 s with a Bandelin Sonopuls HD 2070 sonicator (50%, 20 kHz) to disaggregate the NPs. Final centrifugation at 1200 rpm for 5 min was performed to remove the remaining aggregated NPs and the residual Lig-TA-LTH. 

As a control, NPs containing Lig-TA-LTH but not cobalt were obtained as previously described [39]. Lignin (10 mg/mL) was soaked in a sodium acetate buffer (50 mM, pH 5) solution of 1.5 mg/mL acetosyringone at 50 °C and stirred (250 rpm). Laccase (1% *v*/*v*) was incubated with the lignin solution for 1 h at 50 °C (with stirring applied) to initiate the oxidative reaction. Then, TA (10 mg/mL) and LTH (0.5 mg/mL) were added to the solution, and the reaction proceeded under sonication (20 kHz, 50% amplitude) for 2 h at 50 °C (VCX 750 ultrasonic processor, Sonics, Oklahoma City, OK, USA). Then, the solution was centrifuged at 25,000× *g* for 15 min at 4 °C to remove the residual TA and LTH in the supernatant, and the obtained NPs (Lig NPs) were suspended in water via sonication for 30 s (50%, 20 kHz) and centrifuged at 500× *g* for 5 min to remove the heaviest NPs and the undissolved lignin.

### 2.5. Characterization of Cobalt/Lignin Nanoparticles (CoLig NPs)

#### 2.5.1. Morphological Characterization and Cobalt Loading

The size and size distribution of the CoLig NPs were determined through nanoparticle-tracking analysis (NTA) using a NanoSight LM10 instrument (Malvern Instruments Inc., Malvern, UK) equipped with LM10 Laser (405 nm), NP viewing unit, and sCMOS camera. Several frames of the NP suspension were captured using the NTA 3.2 software to determine the size of the particles. Scanning electron microscopy (SEM) (Merlin Zeiss, Oberkochen, Germany) was used to determine the NPs’ size and morphology. The SEM images were analyzed using ImageJ software (version 1.53t). Dynamic light-scattering (DLS) analyses were performed using a Zetasizer Nano Z (Malvern Instruments Inc., Malvern, UK) to study the size distribution, polydispersity index (PDI), and ζ-potential of the NPs. The content of cobalt in the CoLig NPs was quantified through inductively coupled plasma mass spectrometry (ICP-MS 7800, Agilent Technologies, Santa Clara, CA, USA) calibrated through an internal standard with ^45^Rh and a standard curve of ^59^Co. Before the analysis, the samples were digested with 20% (*v*/*v*) HNO_3_ at 100 °C for 3 h, diluted at a final concentration of HNO_3_ of 2% *v*/*v*, and filtered through a 0.2 μm pore size filter.

#### 2.5.2. MPO and MMP Inhibition Assays

The MPO inhibition capacity of CoLig NPs was analyzed, for which guaiacol was used as a substrate. The samples were dispersed at 1 mg/mL in 0.1 M sodium phosphate buffer, with pH adjusted to 7.5 through the addition of 0.1 M sodium phosphate monobasic. Then, 152 µL of the samples was mixed with 32 µL of MPO (90 U/mL) diluted 1:20 in Milli-Q water and 96 µL of guaiacol solution (16 mM in the buffer previously prepared). The mixtures were incubated at 37 °C for 90 min; then, 190 µL was poured in a 96-well plate, and 10 µL of 10 mM H_2_O_2_ was added to develop the reaction. The absorbance at 476 nm was recorded over time, and the slopes of the curves in the first 150 s were compared. The percentage of MPO inhibition was calculated by assuming that the control values (MPO without NPs) represented 100% activity. The MMP inhibition capacity of CoLig NPs was determined using the Enzcheck Gelatinase/Collagenase Assay Kit and by adapting the protocol to this use. Briefly, 1 mg of the sample was dispersed in 500 µL of buffer (80 mL 1 M Tris HCl buffer pH 7.1 and 20 mL 1 M CaCl_2_). Then, 5 µL of 500 U/mL collagenase type IV from *Clostridium histolyticum* was added to the solution to yield a final concentration of 5 U/mL. The solution was incubated at 37 °C for 24 h. Then, 100 µL of the sample was mixed with 80 µL of buffer and 20 µL of fluorescein conjugate gelatin (100 µg/mL in Milli-Q water). The plate was incubated for 20 min in the dark at room temperature, and fluorescence was measured at λ_ex/em_ = 493/528 nm. The collagenase activity was determined as a percentage of enzyme inhibition compared to the control (a reaction mixture with enzyme and substrate but without NPs).

#### 2.5.3. Antibacterial Activity of CoLig NPs

The susceptibility of *S. aureus* and *P. aeruginosa* to CoLig NPs was determined in 96-well microtiter plates through the broth dilution method as previously described [47]. Different dilutions of CoLig NPs were incubated at 37 °C with bacterial suspensions (5 × 10^5^ CFU/mL) in Nutrient broth. After 24 h, the optical density at 600 nm was measured (Ultramark Microplate Imaging System, Bio-Rad Laboratories S.r.l., Segrate, Italy) to obtain the minimum inhibitory concentration (MIC). As a control, Lig NPs were used.

The morphology of *S. aureus* and *P. aeruginosa* treated with CoLig NPs was examined using SEM (Merlin Zeiss, operating at 1 kV). Bacterial cultures were grown in NB overnight and then diluted to an OD_600_ = 0.01. The suspension was mixed with CoLig NPs to achieve a final concentration of 2.4 mg/mL and transferred to a 48-well plate containing silicon wafers. After 24 h at 37 °C, the liquid was drained, and the bacteria remaining on the wafers were fixed overnight in a 2% paraformaldehyde solution. Finally, the bacteria were dehydrated by incubating the wafers with increasing concentrations of ethanol for 1 h each (25, 50, 75, and 100%). The same process was repeated for untreated bacteria. 

#### 2.5.4. Cytotoxicity of CoLig NPs

The cytotoxicity of CoLig NPs was tested on human cell lines (fibroblasts, BJ-5ta, and keratinocytes, HaCaT) as described previously [47]. Briefly, cells were seeded in a 96-well tissue-culture-treated polystyrene plate (60,000 cells per well) in 100 µL of DMEM supplemented with 200 mM of L-glutamine, 1% penicillin, and 10% (*v*/*v*) fetal bovine serum. After 24 h incubation at 37 °C in a humidified atmosphere with 5% CO2, CoLig NPs dispersed in medium were added at different concentrations. After 24 h, cell viability was determined using the AlamarBlue assay. The medium was discarded, and 100 µL of new medium with AlamarBlue reagent was added (dilution 1:10). After 4 h of incubation, the fluorescence at λ_ex/em_ = 560/590 nm was measured. Cells incubated without CoLig NPs were used as a growth control (100% cell viability).

Cell viability was confirmed through fluorescence microscopy using the Live/Dead Viability/Cytotoxicity Kit (Thermo Fisher Scientific, Waltham, MA, USA). The cells were prepared as previously described and incubated with CoLig NPs at different concentrations for 24 h. Then, the medium was removed, and the cells were stained for 20 min with a PBS solution containing 0.1% *v*/*v* calcein acetoxymethyl and 0.1% *v*/*v* ethidium homodimer-1. The cells were observed using a fluorescence microscope (Nikon/Eclipse Ti-S, The Netherlands) at λ_ex/em_ = 494/517 nm for calcein acetoxymethyl and at λ_ex/em_ = 517/617 nm for ethidium homodimer-1.

### 2.6. Preparation and Characterization of Supramolecular Hydrogel (SM-Gel) Embedded with CoLig NPs

#### 2.6.1. Synthesis of Hydrogel Constituent Polymers

The SM-gel was composed of α-cyclodextrins (αCDs) and a mix of two custom-made poly(ether urethane)s (PEUs), whose acronyms are SHF68 and CHP407. Both these polymers were synthesized and characterized as reported in previous works [32,33]. Briefly, the synthesis process was composed of two steps and conducted under N_2_ flow. In the first step, a pre-polymerization was carried out by reacting macrodiol (P407 or F68 20% *w*/*v* in anhydrous DCE) with HDI (2:1 M ratio with respect to macrodiol) for 2.5 h at 80 °C in the presence of DBTDL catalyst (0.1% wt with respect to macrodiol). In the second step, a chain extender (CDM for P407 or NBoc for F68, 1:1 M ratio with respect to macrodiol, and solubilized in anhydrous DCE at 20% *w*/*v* concentration) was added to the pre-polymer solution, and the reaction was carried out at 60 °C for 90 min. At the end of these two steps, methanol was added to passivate any unreacted isocyanate groups. Then, the PEU solution was precipitated in petroleum ether (4:1 volume ratio with respect to total DCE volume), dried overnight, and purified through solubilization in DCE and precipitation in a diethyl ether–methanol mixture (98:2 *v*/*v*, 5:1 volume ratio with respect to DCE) to remove any unreacted reagent, by-products, and residual catalyst. The obtained materials were kept at 4 °C under vacuum. Furthermore, the PEU containing the NBoc chain extender underwent a de-protection process to remove the Boc caging groups and expose primary amines. The Boc groups’ cleavage was executed through an acid treatment using trifluoroacetic acid (TFA) (as previously reported) [32,33,48]. The acronyms used to identify the synthesized PEUs, CHP407 and SHF68, depended on the reagents used. “C”, “S”, and “H” correspond to CDM, deprotected NBoc, and HDI, respectively, while P407 and F68 refer to the macrodiol used for synthesis. The success of PEU synthesis was assessed using Attenuated Total Reflectance–Fourier Transforme InfraRed (ATR-FTIR) spectroscopy, Proton Nuclear Magnetic Resonance (^1^H NMR) spectroscopy, and Size Exclusion Chromatography, as reported in previous works [32,33,46].

#### 2.6.2. Preparation of the SM-Gel Loaded with CoLig NPs

The SM-gel loaded with CoLig NPs was prepared by adapting the protocol developed by Torchio et al. [33]. Briefly, SHF68 and CHP407 were first solubilized in PBS at 4 °C; then, αCDs at 14% *w*/*v* in PBS with CoLig NPs were added to the polymeric solutions to obtain a final formulation with 3% *w*/*v* poly(urethane), 2.5 mg/mL CoLig NPs, and 10% *w*/*v* αCDs. The samples were vortexed for 20 s, mixed at a volume ratio of CHP407:SHF68 80:20, and then incubated at room temperature to allow gelation. As a control, the same procedure was repeated without the addition of the nanoparticles.

#### 2.6.3. Bovine Serum Albumin (BSA) Uptake in the SM-Gel

The SM-gel’s capacity to absorb BSA was tested to assess the gel’s ability to absorb proteins from aqueous media. The size of BSA size matches the MMPs’ dimensions. The uptake of these enzymes inside the gel can facilitate the action of actives loaded within it. The SM-gels were prepared without the addition of NPs, as previously described, using 10% *w*/*v* αCD concentration and CHP407:SHF68 with a volume ratio of 80:20 and by varying the PEU blend content (1, 3, 5% *w*/*v*). Then, 500 µL of each formulation (1, 3, and 5% *w*/*v*) was poured in vials and allowed to assemble into SM-gels. After 48 h, 500 µL of a solution at 1 mg/mL of BSA in PBS was added. At defined time points (1, 3, 6, 24, 48, and 72 h), the supernatants were collected and analyzed spectrophotometrically at 280 nm to quantify the residual BSA. The results obtained were compared with the initial amount of BSA using a calibration curve obtained from BSA in PBS. After the supernatants’ removal, the hydrogel in the vials was weighed (W_gel,f_), freeze-dried (Telstar LyoAlfa 15, Telstar, Milano, Italy), and weighed again (W_dry_). The solution uptake (BSA uptake%) was calculated using Equation (1), while the hydrogel dissolution (weight loss %) was determined via Equation (2) using a control lyophilized prior to incubation (W_dry,cntr_).
(1)$$\mathrm{BSA}\;\mathrm{uptake}\;\%=\frac{W_{gel,f}-W_{gel,i}}{W_{gel,f}}\times 100$$
(2)$$\mathrm{Weight}\;\mathrm{loss}\;\%=\frac{W_{dry}-W_{dry,cntr}}{W_{dry,cntr}}\times 100$$

#### 2.6.4. Stability of the SM-Gel Loaded with CoLig NPs 

The SM-gel was prepared at a 3% *w*/*v* PEU concentration, as described previously; then, 500 µL of the solution was poured into vials prior to gelation and incubated for 48 h at room temperature to allow for the complete development of the SM-gel network. Afterward, the vials were weighed (W_gel,i_), 500 µL of phosphate buffer saline (PBS) was added, and the vials were incubated at 37 °C. At defined time points (6, 24, 48, and 72 h), the samples were collected, weighed after the removal of residual PBS (W_gel,f_), lyophilized, and weighed again (W_dry_). In contrast with what was performed previously, in the BSA uptake test, at each time point, the PBS was renewed for all samples to simulate exudate renewal. PBS uptake (PBS uptake%) was calculated using Equation (3), while hydrogel dissolution (weight loss %) was obtained from Equation (2) using a control lyophilized prior incubation with PBS (W_dry,cntr_). Analyses were conducted in triplicate, and results are reported as mean ± standard deviation. The SM-gel without CoLig NPs was tested as a control.
(3)$$\mathrm{PBS}\;\mathrm{uptake}\;\%=\frac{W_{gel,f}-W_{gel,i}}{W_{gel,f}}\times 100$$

#### 2.6.5. Rheological Tests

The mechanical properties of the SM-gels (3% *w*/*v* PEU concentration, with and without NPs) were studied through rheological characterization using a stress-controlled rheometer (MCR302, Anton Paar GmbH, Graz, Austria) equipped with a 25 mm parallel plate configuration. The hydrogel was prepared as described previously and deposited into a syringe prior to gelation to allow for an easier loading on the rheological plate. Around 0.4 mL of SM-gel was used in each experiment; the sample thickness was set to 0.6 mm, and the value of normal force was set to 0 N. Strain sweep tests were carried out at 37 °C at an angular frequency of 1 rad s^−1^ with a strain range of 0.01–500%. For each sample, the test was repeated after 15 min to assess the ability of the hydrogels to recover the mechanical properties. Frequency sweep tests were performed within the linear viscoelastic region (i.e., 0.1% strain) at 25 and 37 °C with angular frequencies between 100 and 0.1 rad s^−1^. To study the self-healing ability of SM-gels, self-healing strain tests were performed at 1 Hz frequency with the application of cyclical deformations. The cycles were composed of a recovery phase with low deformation (0.1%, 120 s) and a rupture phase with high deformation (100%, 60 s). After three cycles, the initial recovery phase was applied again to evaluate the residual mechanical properties.

#### 2.6.6. CryoSEM

The SM-gels with and without CoLig NPs were prepared as previously described and placed in a syringe prior to gelation to enable easier loading on the cryoSEM stubs. The samples were analyzed using a Hitachi S-3500N scanning electron microscope (Hitachi High-Tech Co., Tokyo, Japan) at the Institute of Marine Sciences of the Spanish Research Council. The gels were mounted on aluminum stubs and plunged into liquid nitrogen slush. Once the materials were frozen, they were transferred under vacuum conditions to a cryo-preparation chamber, i.e., the Quorum PP3000T (Quorum Technologies, Ltd., Lewes, UK). The preparation chamber was under high vacuum and fitted with a cold stage where the samples were cold-fractured, sublimed at −90 °C for 4 min, sputter-coated with platinum, and transferred to a cold stage in the chamber of the microscope. The samples were maintained at −130 °C during observation at an acceleration voltage of 5 kV. The number of branches indicating the level of crosslinking was obtained using the Skeleton tool of ImageJ.

#### 2.6.7. CoLig NPs Interaction with SHF68

The interaction between CoLig NPs and SHF68 was studied using ATR-FTIR spectroscopy. Briefly, 5 mg of SHF68 was solubilized in 500 µL of distilled water and mixed with 5 mg of CoLig NPs, which was also suspended in 500 µL of distilled water. The mix was allowed to interact at 37 °C overnight and was then freeze-dried. The sample was finally analyzed using ATR-FTIR spectroscopy, as previously described.

#### 2.6.8. CoLig NPs’ Release from the SM-Gel

The SM-gel (3% *w*/*v* polymer concentration) containing CoLig NPs was prepared as described previously; then, 500 µL of the solution was poured into vials and incubated for 48 h at room temperature to allow for the complete development of the SM-gel network. Afterward, 500 µL of PBS was added to each vial and incubated at 37 °C. At each time point (6, 24, 48, and 72 h), the supernatant was collected, the volume was measured, and fresh PBS was added. At the 72 h mark, the SM-gel was completely dissolved. The release solutions collected were analyzed using ICP-MS and fluorescence analysis to quantify the amount of Co and lignin present, respectively. For the determination of the Co content, the samples were prepared for ICP-MS analysis, as previously described. For the fluorescence analysis, 100 µL of the release solution was poured into a 96-well plate, and fluorescence was measured at λ_ex/em_ = 480/610 nm. The calibration curve was determined using CoLig NPs. All samples were measured in triplicate, and results are reported as mean ± standard deviation. The release rate was calculated using Equation (4)
(4)$$\mathrm{Release}\;\mathrm{rate}=\frac{M_{i}}{\Delta t}$$
where *M_i_* is the quantity of CoLig NPs released in mg using the fluorescence analysis at a defined step *I* and Δ*i* is the duration of the time step (h). 

To study the kinetics of release, the Korsmeyer–Peppas model (Power law) [49] was applied to the release data collected within the first 24 h. In detail, Equation (5) was implemented to calculate the release exponent n: (5)$$\frac{M_{t}}{M_{\infty }}=Kt^{n};\;\mathrm{Log}(\frac{M_{t}}{M_{\infty }})=n\mathrm{Log}(t)+\mathrm{Log}(K)$$
where *M_t_* is the mass of released CoLig NPs at a defined time step *t*, *M*_∞_ is the total encapsulated payload, and K is a constant of incorporation of structural modification (also known as the release velocity constant). In cylindrical samples, *n* values of 0.45 and 0.89 are related to release mechanisms based on diffusion or swelling, respectively. Values ranging between 0.45 and 0.89 correspond to anomalous transport, which is a combination of the abovementioned release phenomena. *n* values greater than 0.89 correspond to a mode of transport in which tensions within the hydrogel network are generated via solvent uptake and relevant mass exchange with the external environment.

#### 2.6.9. Antibacterial Activity of the SM-Gel Containing CoLig NPs

The antibacterial activity of the SM-gel containing CoLig NPs was assessed through the morphological evaluation of *S. aureus* and *P. aeruginosa* via SEM. A total of 500 mg of SM-gel loaded with CoLig NPs was prepared in sterile glass vials. The process was performed using sterile PBS under a biological hood to ensure the sterility of the gel. Subsequently, 0.5 mL of NB was poured into the vials such that it contacted the SM-gel either with or without CoLig NPs. The samples were incubated for 24 h at 37 °C, and the hydrogel extracts were collected. In parallel, bacterial cultures were grown in NB overnight and then diluted to an OD_600_ = 0.01. The suspension was mixed with the hydrogel extract and transferred to a 48-well plate containing silicon wafers. After 24 h at 37 °C, the liquid was drained, and the bacteria remaining on the wafers were fixed overnight in a 2% paraformaldehyde solution. Finally, the bacteria were dehydrated by incubating the wafers with increasing concentrations of ethanol for 1 h each (25, 50, 75, and 100%). The same process was repeated for untreated bacteria. The samples were analyzed through SEM, as previously described.

#### 2.6.10. Cytocompatibility of the SM-Gel Containing CoLig NPs

The SM-gel’s cytocompatibility was evaluated according to ISO 10993 recommendations. A total of 100 mg of SM-gel loaded with CoLig NPs was prepared in sterile glass vials. This process was performed using sterile PBS under a biological hood to ensure the sterility of the gel. Subsequently, 1 mL of cell culture medium was poured such that it contacted the hydrogel. The samples were incubated for 24 h at 37 °C, and the hydrogel extracts were collected. The process was repeated for SM-gel without CoLig NPs as a control. Simultaneously, BJ-5tα and HaCaT were cultivated for 24 h in the same medium as previously described. Then, 100 μL of the collected samples was added to each well of the cultured cells and incubated for 24 h. Cell viability was determined using the AlamarBlue test and the Live/dead assay as previously described.

### 2.7. Statistical Analysis

Results are reported as mean ± standard deviation. Statistical analysis was performed through JASP for Windows 10 (version 0.9.2). To compare results, one-way ANOVA analysis was performed, followed by Bonferroni’s multiple comparison test. Statistical significance was assessed as follows: n.d.—non-significant difference, * *p* < 0.05, ** *p* < 0.01, and *** *p* < 0.001.

## 3. Results and Discussion

With the increase in the lifespan and the progressive aging of the global population, the incidence of CW is projected to increase in the coming years. Therefore, the development of new and more effective treatments for these pathologies is in high demand in order to reduce their burden on patients and the health care system. This study developed an SM-hydrogel-based drug delivery system that can be easily applied to the wound bed and delivers anti-inflammatory and antibacterial CoLig NPs. 

### 3.1. CoLig NPs’ Synthesis and Characterization

To improve lignin’s reactivity and its ability to inhibit both MPO and MMPs, it was conjugated with TA, a natural polyphenol, and LTH, an MMP inhibitor, in an enzymatic reaction catalyzed by laccase, yielding Lig-TA-LTH. TA was added to increase phenolic content and thus improve lignin’s interfacial reactivity and antioxidant and antimicrobial properties. Moreover, TA has shown inhibitory effects against MPO [50,51], which is largely overexpressed in CWs. On the other hand, the hydroxamic group of LTH should specifically inhibit Zn-containing MMPs [52]. The increase in phenolic and ammino content verified that the conjugation was successful. The phenolated lignin presented 0.15 ± 0.007 mg GAE per mg of conjugate, which can be compared to the initial 0.12 ± 0.002 mg GAE per mg of pristine lignin (*p* = 0.013). The number of primary amines present in Lig-TA-LTH was 2.4 ± 0.2 µg LTH per mg of conjugate, confirming the successful grafting of LTH. The Lig-TA-LTH ATR-FTIR spectrum in Figure 1 shows the presence of a peak around 1700 cm^−1^ related to the C=O stretching vibration of unconjugated carbonyl groups [53]. The increase in intensity in this peak compared to the small shoulder observed in pristine lignin confirmed the presence of TA in Lig-TA-LTH. Moreover, in Lig-TA-LTH, a shift of the band at 3000–3600 cm^−1^ was observed, which can be ascribed to the stretching of the OH- bond on the aromatic ring [54,55]. This could prove that the cross-linking reaction of lignin and tannic acid occurred during laccase-assisted phenolation [39]. 

The conjugated lignin was then used as a reducing agent to synthesize CoLig NPs. During the process, a change was observed in the solution color from light to dark brown. The presence of Co in the NPs was confirmed through ICP-MS and measured to be 41.6 ± 0.2 µg per mg of CoLig NPs. In addition, the phenolic content decreased from 0.15 ± 0.007 to 0.11 ± 0.005 mg GAE per mg of material (*p* = 0.001), thereby confirming cobalt’s reduction. 

The CoLig NPs were further characterized via DLS, NTA and SEM. The size of the CoLig NPs varied depending on the technique used to observe them. Table 1 is a summary of the values obtained. As expected, SEM analysis yielded the smallest diameter since it was used to analyze the dried NPs, while both DLS and NTA were used to measure hydrodynamic diameter. The polydispersity of the NPs measured via DLS was 0.22 ± 0.01, and their surface charge (ζ potential) was -23.1 ± 0.8 mV, demonstrating colloidal stability (Table 1). Finally, the SEM images revealed an irregular round shape of the CoLig NPs, which was probably the result of the formation of compact clusters after the interaction between Co and Lig-TA-LTH (Figure 2). The observed morphology is similar to that of tellurium–lignin NPs [47] and silver–lignin NPs [36] obtained through a similar process in previous works.

### 3.2. MPO and MMP Inhibitory Capacity of CoLig NPs

The CoLig NPs were tested to assess their ability to inhibit MPO and MMPs, for which high inhibitory capacity (>80%) was revealed in both cases with statistically significant differences when compared with lignin (*p* < 0.001) (Figure 3). These enzymes are usually overexpressed in CWs and contribute to stalling the wound-healing process in the inflammatory phase of healing. MPO catalyzes the oxidation of substrates producing oxidizing and halogenating agents that promote the degradation of healthy tissue [56]. Natural compounds presenting phenols can act as scavengers of radical species by means of the hydroxyl groups acting as H-donors [57]. Additionally, they can compete as a substrate for MPO and inhibit HOCl production [58]. MPPs, on the other hand, are a family of Zn-containing peptidases involved in ECM degradation [14,52]. Hydroxamic groups are known to inhibit MMPs by strongly interacting with the Zn(II) in the active site of these enzymes [52]. Previous studies also showed some zinc-chelating activity of TA [58]. In this work, the MPO inhibitory capacity of CoLig NPs can be acribed to the increased phenolic content of lignin [51], while the inhibition of MMPs can be ascribed to the synergic activity of LTH and TA. Interestingly, the inhibitory activity of the nano-formulation was higher than that observed for Lig-TA-LTH alone before reduction with cobalt (*p* < 0.01 for MPO and *p* < 0.001 for MMPs). This could be related to the presence of cobalt, which has been shown to interact with proteins [59] and may act at the level of the Zn active center of MMPs. Additionally, the nanoscopic size of the particles could also improve the interaction of the NPs’ active groups with the surrounding environment, thereby facilitating binding with enzymes [60]. 

### 3.3. CoLig NPs’ Antibacterial Properties and Cytotoxicity

The CoLig NPs’ antibacterial properties were assessed with respect to two common pathogens found in CWs: the Gram-negative *P. aeruginosa* and the Gram-positive *S. aureus* [61,62]. The MIC values obtained were 1.2 mg/mL and 2.4 mg/mL for *S. aureus* and *P. aeruginosa,* respectively (Table 2). For the Lig NPs, based on Lig-TA-LTH without the presence of cobalt, the MIC value obtained was 3.3 mg/mL for both bacterial strains. This test demonstrated that the presence of cobalt enhanced the antibacterial activity of the CoLig NPs. 

The difference in susceptibility between Gram-positive and Gram-negative species can be ascribed to the different components of the bacterial cell wall. Indeed, the CoLig NPs probably exerted their antibacterial effect by damaging the cell membrane. This type of interaction was also observed in previous studies analyzing the antibacterial mechanism of lignin and lignin nanoparticles [39,63]. To confirm that this interaction has occurred, SEM analysis of bacteria treated with CoLig NPs was performed to study cell morphology. After treatment with NPs, both *S. aureus* and *P. aeruginosa* cells showed irregular and wrinkled surfaces with some depressed areas. On the contrary, the control cells presented regular and smooth surfaces (Figure 4). These results confirmed the hypothesized mechanism of action of CoLig NPs based on the membrane–NPs interaction.

The potential cytotoxicity of the CoLig NPs was tested with respect to two human cell lines present in the human skin: fibroblasts (BJ-5ta) and keratinocytes (HaCaT). The % of cell viability at 0.6 mg/mL and below was higher than 70% for both cell lines (*p* < 0.001 when comparing treatment with 0.6 and 1.2 mg/mL) (Figure 5). The viability results were confirmed through a Live/Dead assay (Appendix A), which is in accordance with previous works [64,65]. The observed cytotoxicity can be mainly ascribed to the presence of cobalt since the Lig NPs only presented toxicity at concentrations above 1.7 mg/mL (Appendix A). Nevertheless, cobalt’s reduction in the CoLig NPs decreased cobalt cytotoxicity. In fact, the toxicity of the cobalt ions and cobalt oxide NPs was observed at concentrations ranging between 5 and 25 µg/mL [66,67], while in the CoLig NPs at the cytocompatible concentration of 0.6 mg/mL, the cobalt content was 24.5 µg/mL. The CC_50_ (concentration resulting in 50% reduction in cell viability) values were 0.9 mg/mL and 1.1 mg/mL for HaCaT and BJ-5ta, respectively. Even though the CC_50_ values are lower than the MIC, it is possible to identify a CoLig NPs concentration (0.6 mg/mL) capable of highly inhibiting *S. aureus* growth while maintaining cytocompatibility (Figure 6 and Appendix A). Nevertheless, future studies will focus on improving the system’s cytocompatibility using coatings targeting proteins specific to the bacterial membranes, thus minimizing interaction with healthy cells.

### 3.4. Design of the Nano-Enabled SM-Gel

The SM-gel is composed of αCDs and a blend of two PEUs (CHP407 and SHF68) that were previously synthesized and characterized [32,33,46,48]. The SM network has been thoroughly studied [32,33], highlighting its potential as a drug delivery system. It has shown injectability and self-healing properties due to the physical nature of the interactions between the hydrogel’s components [32,33]. In the SM-gel, gelation is driven by the formation of PPRs due to the host–guest interactions of αCDs with the PEO blocks present in the PEU chains. Additionally, the PEUs are composed of amphiphilic Poloxamers^®^ (P407 and F68) acting as macro-diols, which form micelles in an aqueous environment, thereby further facilitating the assembly of the SM-gel. The advantage of using a blend of two PEUs in this hydrogel lies in the possibility to combine the different properties that these materials have shown [32,33,48]. CHP407 is based on P407 and presents CDM as a chain extender. The linearity of its chain, without the presence of pending groups, favors interaction with αCDs, resulting in a hydrogel with good stability and mechanical properties. SHF68 is synthesized using F68 as macro-diol and NBoc as chain extender. A subsequent deprotection process eliminates the Boc group of the chain extender and exposes primary amines. Even though the hydrogel obtained using SHF68–αCDs interaction presents reduced mechanical properties, the presence of amines can facilitate reaction with several groups and can be exploited for the further functionalization of the material. Additionally, the higher solubility of SHF68 in water facilitates interactions with aqueous media and material exchange with the surrounding environment. These characteristics endow SHF68-based gels with higher responsiveness compared to the CHP407-based ones. 

In this work, a blend of CHP407 and SHF68 at volume ratio of 80:20 was selected as the best compromise between mechanical behavior and responsiveness. First, SM-gels formed in PBS at 10% *w*/*v* αCDs and at different polymeric concentrations (1%, 3%, and 5% *w*/*v*) were studied to select the optimal conditions for the envisaged application. A previous study conducted by our group [33] investigated the SM-gel’s stability and PBS uptake behavior at different PEU concentrations. In the SM-gels, PBS uptake was minimal, while a reduction in dry weight proved that the network components were progressively released from the hydrogels. Therefore, it was inferred that an exchange of materials with the surrounding environment took place, which progressively eroded the gel. However, it was observed that the hydrogels maintained their integrity even after losing more than 50% of their constituents. 

In this work, the SM-gels were further characterized to ascertain their ability to absorb proteins from the aqueous environment. This capacity can be helpful in the field of CW healing since this pathology presents exudates rich in components such as MMPs that facilitate the prolongation of the inflammatory state [68,69]. Therefore, the correct management of wound exudate is crucial to the healing process. Moreover, the uptake of these components inside the hydrogel network could facilitate the interaction with loaded actives. In this experiment, BSA was used as a model protein since its size (66 kDa) matches the range of the MMPs’ dimensions (50–70 kDa) [70]. Throughout the experiment, the hydrogel presented minimal PBS uptake, and the volume of the solution extracted did not vary significantly. Nevertheless, a progressive reduction in hydrogel constituents was observed through the measurement of the hydrogel’s dry weight (Figure 7a). The % of BSA absorbed inside the gels also presented a similar behavior and increased (Figure 7b) until reaching a plateau after 72 h. These results further prove that an exchange of material from and to the hydrogels was taking place, as previously suggested [32,33]. Moreover, the % of BSA absorbed depended on the PEU concentration, and the differences between the three conditions were statistically relevant (*p* < 0.001). Interestingly, the highest uptake was observed for the hydrogel at a 1% *w*/*v* PEU concentration (around 50%), while at 5% *w*/*v* PEU content, only 30% of the BSA was absorbed. This can be explained by the peculiarity of these hydrogel systems. In fact, previous studies [33] showed that at low polymer concentrations, the networks of SM-gels present a more marked SM character, which results in their greater ability to maintain their shape, while strongly interacting with the surrounding environment. This is probably the reason behind the improved capacity to absorb BSA for the gel with 1% *w*/*v* PEU content. Additionally, a lower PEU concentration should minimize the risk of possible cytotoxic effects.

Nevertheless, a high polymer concentration allows the hydrogel to maintain its integrity for longer periods. Therefore, a compromise between responsivity and stability is required. In this work, the condition selected was 3% *w*/*v* PEU content. This SM-gel was loaded with the CoLig NPs previously obtained, and the behavior of the nano-enabled system was further characterized. 

### 3.5. CoLig NPs’ Integration in the SM-Gel and Characterization

The CoLig NPs were loaded at 2.5 mg/mL in the SM-gel (3% *w*/*v* polymer concentration, CHP407:SHF68 volume ratio of 80:20, and 10% *w*/*v* αCDs), and the resulting hybrid hydrogels were subjected to stability tests, rheological characterization, and cryoSEM imaging. The result of the CoLig NPs’ integration was a homogeneous hydrogel with a brown coloration, which was related to the presence of lignin (Appendix A). This concentration was selected to match the MIC values. The potential cytotoxicity of the released NPs depends on their release rate and will be discussed in the following sections. Nevertheless, tests aiming to characterize the system’s antibacterial activity and cytotoxicity were also performed. The results of the stability tests are in accordance with those observed in a previous study [33]. The hydrogel did not present PBS uptake (Appendix A) during the test, apart from the very last time point in which it started losing its integrity. However, the SM-gel’s dry weight was progressively reduced until it was completely dissolved. This suggests that the hydrogel was indeed exchanging material with the exterior, but it was able to maintain a stable network even when most of its content was released. Notably, the addition of CoLig NPs increased the SM-gel’s stability (Figure 8), which was probably due to interactions of the phenolic shell of the NPs with SM-gel components, such as the primary amines exposed by SHF68 and αCDs.

Rheological tests also showed some differences between the unloaded SM-gel and the one loaded with NPs. Rheological testing is usually performed to assess the mechanical properties of a hydrogel, which, in turn, determine its processability and possible applications. A frequency sweep test was performed to evaluate the degree of SM-gel development. Both the unloaded SM-gel and the one loaded with CoLig NPs presented the characteristics of a fully developed gel (Figure 9a). For all tested frequencies, the storage modulus (G′) was greater than the loss modulus (G′′), and these values were independent over an angular frequency. Nevertheless, the SM-gel loaded with CoLig NPs showed improved mechanical properties, with a G′ value at 10 rad/s of 6300 Pa compared to a G′ value of 4965 Pa for the unaltered SM-gel. An amplitude sweep test was performed to define the linear viscoelastic region (LVE) and the flow point (T_f_) at which the G′ and G′′ curves cross, during which the hydrogel started behaving as a viscous liquid due to the mechanical breakage of its network. The SM-gel loaded with NPs showed higher storage and loss moduli values within the LVE, i.e., 7114 Pam and 862 Pa, respectively, but lower strain corresponding to the flow point, namely, 13.6% (Figure 9b and Table 3). The integration of NPs in the SM-gel resulted in the stiffening of the system in accordance with the results from the frequency sweep test [71,72]. These results strengthen the hypothesis of an interaction between phenolic groups exposed by the NPs and hydrogel components (primary amines from SHF68 and αCDs), as claimed when observing the improved stability of the SM-gel.

The self-healing capacity of the hydrogel was not affected by the presence of the CoLig NPs, as observed in the time-dependent strain test (Appendix A). The percentages of G′ recovery after three rupture cycles were 87% and 89% for the SM-gel loaded with and without CoLig NPs, respectively (Table 3). 

CryoSEM is a technique that requires a sample to be frozen at cryogenic temperatures and imaged under a high vacuum. This process reduces damage related to drying, and it has been applied to study the structure of sensitive samples, such as hydrogels [73,74]. The SM-gel with CoLig NPs showed a more branched structure compared to the unaltered SM-gel (Figure 10). The reason for the increased degree of crosslinking could be related to the synergic coexistence of different processes. It was theorized that electrostatic interactions could occur between the highly reactive phenolic groups present on the CoLig NPs and the primary amines exposed by SHF68 in the SM-gel. To test this hypothesis, the CoLig NPs and SHF68 were mixed and left to interact overnight at 37 °C. Then, their interaction was analyzed through ATR-FTIR spectroscopy. The spectra of the SHF68-CoLig NPs mix showed a shift in the broad peak of around 3400 cm^−1^ from 3320 cm^−1^ to 3360 cm^−1^. This peak is related to the stretching of the OH- bond on the aromatic ring and may indicate interactions of the phenolic groups on CoLig NPs with the polymer (Figure 11). Additionally, cyclodextrins are also known to form inclusion complexes with phenols [75]. Therefore, the free αCDs present in the SM-gel could interact with the CoLig NPs promoting the hydrogel’s crosslinking. Finally, interactions between aromatic groups present on the CoLig NPs may also favor the network formation of the hydrogel [76]. 

### 3.6. CoLig NPs’ Release from the SM-Gel

The CoLig NPs’ release from the hydrogel was studied through two independent tests: ICP and fluorescence analysis. The ICP test allowed for the quantification of the cobalt released from the gel, while the fluorescence analysis at 480/610 nm revealed the amount of released lignin. The results showed linear release kinetics for both analyses until 24 h (release rate = 0.007 mg CoLig NPs/h); then, the release rate increased to 0.04 mg CoLig NPs/h (Figure 12). These release kinetics matched the SM-gel dissolution trend determined through the stability test (Figure 8), suggesting that the release of the cargo from the SM-gel is driven by its progressive dissolution. This was also confirmed using the Korsmeyer–Peppas model, which describes drug release from polymeric systems. Indeed, after fitting the results of the first 24 h, the release exponents (n) obtained were 0.6 and 0.87 for the ICP and fluorescence analyses, respectively. This indicates a non-Fickian form of transport in which the drug is released through the combined effects of diffusion and the matrix’s progressive erosion [49,77]. The *n* values obtained indicate a release in close correspondence with zero-order kinetics (*n* = 0.89), which is a highly desirable property in drug-releasing systems since it corresponds to a constant release rate over time and the avoidance of burst release that could affect a system’s cytocompatibility. Additionally, at 24 h, the SM-gel released around 17% of its payload, corresponding to 0.4 mg/mL of the CoLig NPs. This concentration should grant cytocompatibility, but further tests were performed to confirm this. Finally, the accordance between the two analyses confirmed the preserved integrity of the CoLig NPs. This is proof that the CoLig NPs inside the SM-gel maintained their integrity and did not release Co ions in an uncontrolled way, which might have resulted in cytotoxic effects.

### 3.7. Antibacterial Activity and Cytocompatibility of the SM-Gel Loaded with CoLig NPs

The antibacterial activity of the SM-gel loaded with CoLig NPs was confirmed according to the change in the morphology of *S. aureus* and *P. aeruginosa* upon incubation with the gel extract collected at 24 h. This approach allowed for the correlation of the antimicrobial action with the CoLig NP release rate observed in previous tests. The concentration of CoLig NPs in the gel extract was estimated to be around 0.5 mg/mL, which is lower than the MIC values obtained for both bacteria strains. Nevertheless, the 24 h collected extracts from the SM-gels with CoLig NPs were able to alter the bacterial surface (Figure 13), confirming the previously observed mechanism of action of CoLig NPs through interaction with the bacteria membrane. This was especially evident in the *P. aeruginosa* cells, in which the membrane presented a clearly irregular and wrinkled surface. In the sample of *S. aureus* treated with the SM-gel with CoLig NPs, a lower number of cells presenting an irregular membrane was observed. This could be related to dead and dying cells’ reduced adhesion to surfaces, which makes them difficult to image. Indeed, fewer cells are normally present on the wafer of treated bacteria compared to controls. 

Interestingly, the unaltered SM-gel was also capable of partially inducing surface modifications. This is probably due to the amphiphilic nature of the polyurethanes that may interact with the bacterial membrane, as suggested in a previous work [78]. This synergic action of the SM-gel and CoLig NPs improved the antibacterial activity of the system. Indeed, antibacterial activity was clearly present even when the CoLig NP concentration in the gel extracts was lower than the MIC. Therefore, combining CoLig NPs and SM-gels helps boost antibacterial properties.

Finally, the cytocompatibility of the SM-gel loaded with CoLig NPs was evaluated against BJ-5ta and HaCaT. The % of cell viability for the SM-gel with and without CoLig NPs was higher than 70% for both cell lines (Figure 14). The viability results were further confirmed through a Live/Dead assay (Appendix A). These results cast the developed formulation as a highly promising solution for the envisaged application, evidencing that the system presents antibacterial action while maintaining cytocompatibility.

## 4. Conclusions

This work presents a promising system for treating chronic wounds that combines the beneficial properties of SM hydrogels with the action of dual-functional lignin–cobalt NPs capable of mitigating an inflammatory state while reducing bacterial burden. 

The CoLig NPs were successfully synthesized through a green procedure. They were capable of regulating MPO and MMPs’ activities and presented bacterial inhibition, especially against the Gram-positive *S. aureus*, at cytocompatible antimicrobial concentrations. Ongoing research is focusing on improving the system’s antibacterial capacity while further reducing its cytotoxicity.

In parallel, the characteristics of the SM-gel were finely tuned to adapt it to CW treatment. The host–guest interactions of αCDs with PEUs were exploited to form PPRs and promote gelation at low polymeric concentrations, thereby reducing the risk of cytotoxic effects. By measuring the SM-gel’s capacity to absorb BSA, it was confirmed that the gel can interact with the surrounding environment while maintaining its shape. This ability to exchange materials with aqueous media is a desirable property in drug delivery systems for treating CWs. The management of exudate is essential for promoting healing, and the uptake of proteins and enzymes from exudate can facilitate their interaction with the actives loaded inside the SM-gel. In this work, we have also demonstrated that the reactive phenols exposed by CoLig NPs can interact with the SM-gel components, mainly primary amines of SHF68, resulting in improved mechanical properties and increased stability of the hydrogel. Furthermore, the CoLig NPs were released from the optimized SM-gel with close to zero-order kinetics, which is a highly desirable property in smart drug delivery systems. 

In summary, the lignin–cobalt nano-enabled SM hydrogel proposed herein can absorb proteins, mitigate inflammatory states, and exert antimicrobial action. The injectability of the hydrogels would allow for their applicability to wounds of any shape. Further research is required to better characterize the system and test its efficacy and safety in a more relevant environment mirroring in vivo conditions. Nevertheless, the system is a highly promising and versatile candidate for the treatment of CWs. Additionally, this work investigated the possibility of tuning the properties of SM hydrogels by selecting appropriate PEU concentrations, exploiting the combination of different PEUs, and exploiting the interaction between reactive groups present on NPs and polymer backbones. These findings constitute an important body of information for the design of effective drug delivery systems. 

## Figures and Tables

**Figure 1 pharmaceutics-15-01717-f001:**
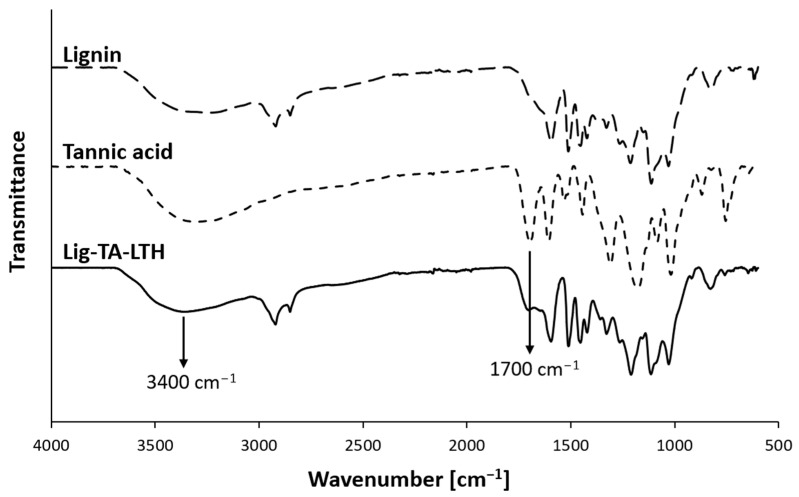
Comparison of ATR-FTIR spectra of lignin, tannic acid, and lignin conjugated with TA and LTH (Lig-TA-LTH). The arrows indicate relevant peaks proving the conjugation of TA with lignin.

**Figure 2 pharmaceutics-15-01717-f002:**
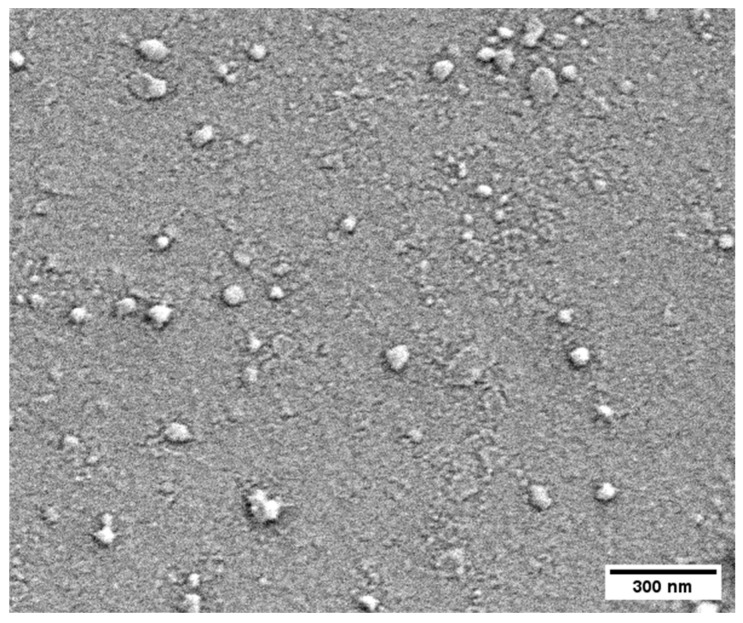
CoLig NPs’ morphology as assessed through SEM imaging. The CoLig NPs have an irregular round shape that is the result of cluster formation after Co and Lig-TA-LTH interaction.

**Figure 3 pharmaceutics-15-01717-f003:**
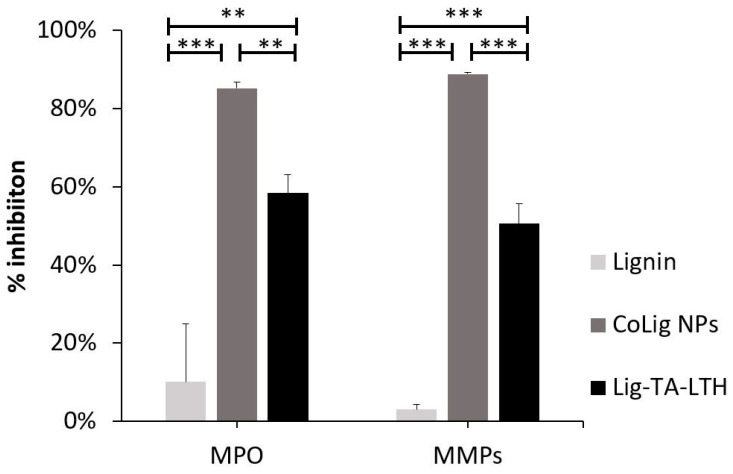
MPO and MMP inhibition of CoLig NPs. *** refers to *p* < 0.001 and ** to *p* < 0.01.

**Figure 4 pharmaceutics-15-01717-f004:**
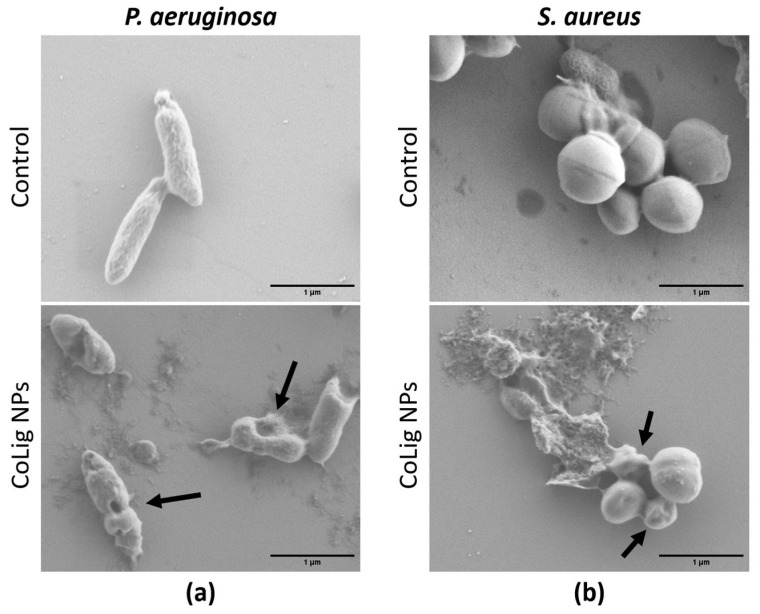
SEM images of bacteria after treatment with CoLig NPs. The bacteria species usead are (**a**) *P. aeruginosa* and (**b**) *S. aureus*. Arrows indicate cells with damaged membranes.

**Figure 5 pharmaceutics-15-01717-f005:**
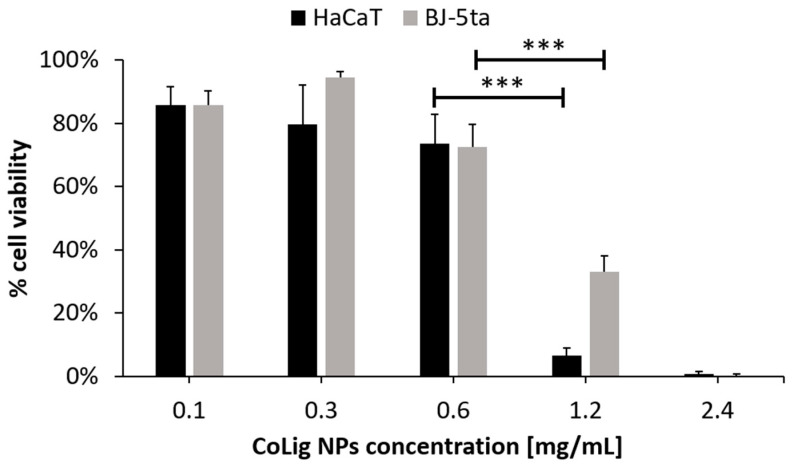
Cytotoxicity of CoLig NPs against human fibroblasts (BJ-5ta) and keratinocytes (HaCaT). *** refers to *p* < 0.001.

**Figure 6 pharmaceutics-15-01717-f006:**
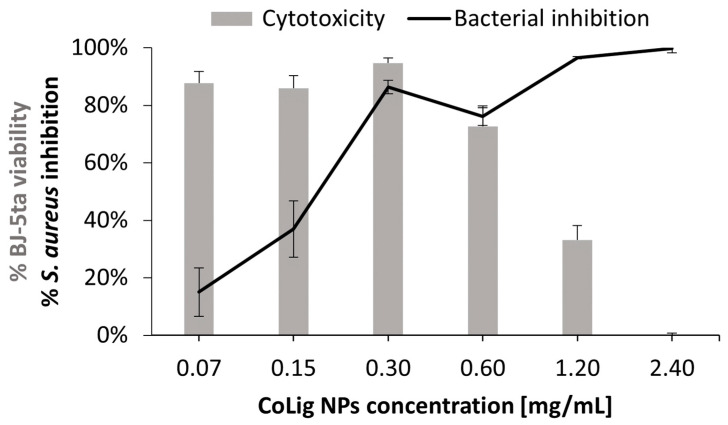
Comparison between the CoLig NPs’ cytotoxicity toward BJ-5ta and inhibitory capacity toward *S. aureus*.

**Figure 7 pharmaceutics-15-01717-f007:**
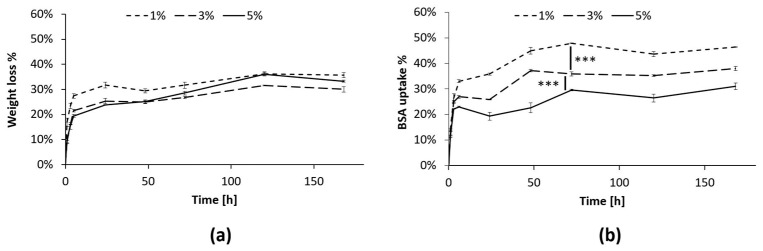
BSA uptake test in the SM-gels with a CHP407:SHF68 volume ratio of 80:20, an αCDs concentration of 10% *w*/*v*, and different overall polymeric concentrations (1, 3, and 5% *w*/*v*). (**a**) % of dry weight reduction in the SM-gels at 1, 3, and 5% *w*/*v* polymer concentrations. (**b**) % of BSA absorbed in the SM-gels from the aqueous medium. After 72 h, the differences between the three conditions remained stable, presenting statistically relevant differences. *** refers to *p* < 0.001.

**Figure 8 pharmaceutics-15-01717-f008:**
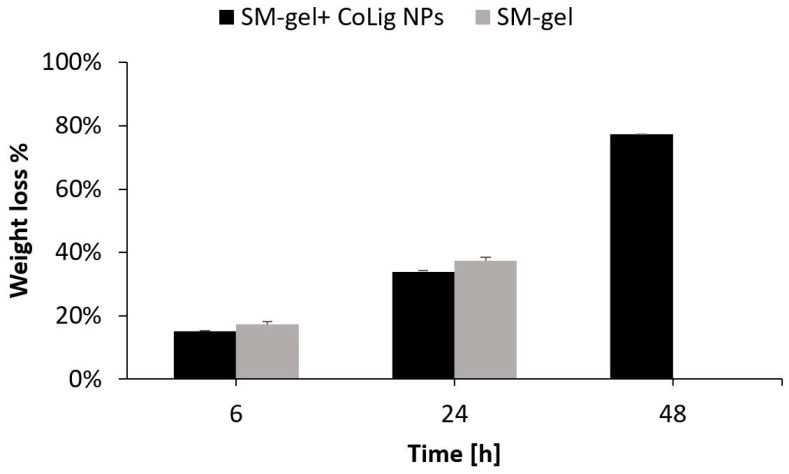
Stability in PBS of the SM-gel loaded with CoLig NPs. An unaltered SM-gel was also tested as control condition.

**Figure 9 pharmaceutics-15-01717-f009:**
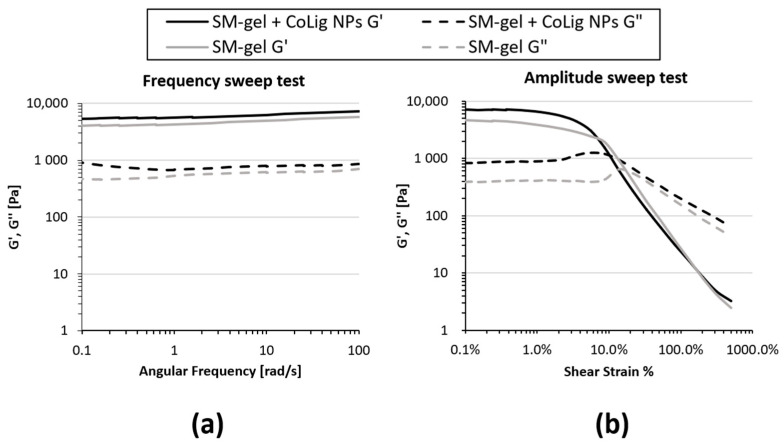
Rheological properties of the SM-gel loaded with CoLig NPs. An unaltered SM-gel was also characterized for comparison. (**a**) Frequency sweep test and (**b**) amplitude sweep test. The addition of CoLig NPs increased the system’s rigidity.

**Figure 10 pharmaceutics-15-01717-f010:**
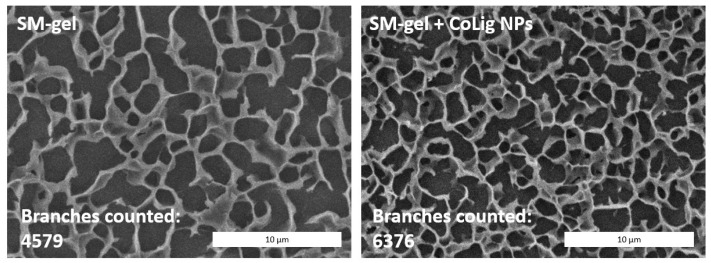
CryoSEM images of the SM-gel with or without CoLig NPs. The addition of CoLig NPs increased the SM-gel’s degree of crosslinking. The numbers of branches of the SM-gels were calculated using the Skeleton tool of ImageJ.

**Figure 11 pharmaceutics-15-01717-f011:**
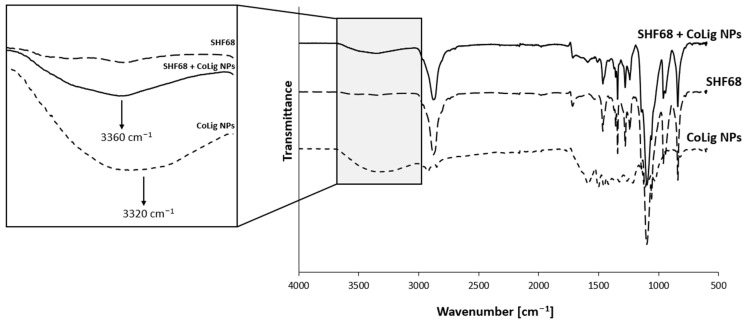
Comparison of ATR-FTIR spectra of SHF68, CoLig NPs, and their combination. The arrows indicate the peak shift proving the interaction between CoLig NPs and SHF68 chains.

**Figure 12 pharmaceutics-15-01717-f012:**
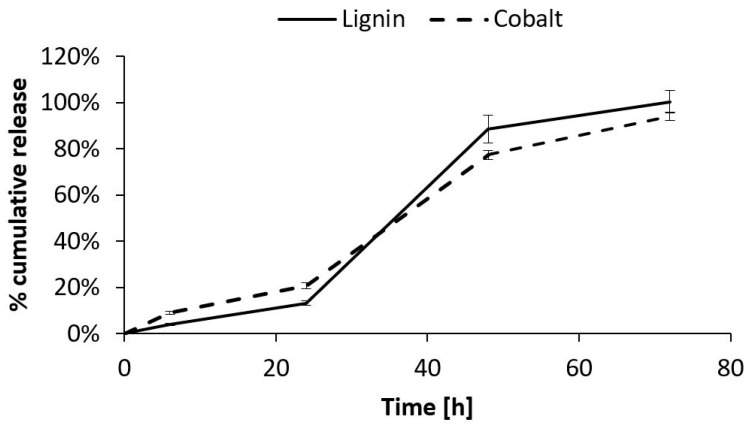
Quantification of CoLig NPs’ release from SM-gel through two different methods: ICP and fluorescence analysis. The CoLig NPs’ release appeared to be close to the 0-order and matched SM-gel dissolution. Moreover, the two analysis methods were in accordance, confirming the NPs’ integrity.

**Figure 13 pharmaceutics-15-01717-f013:**
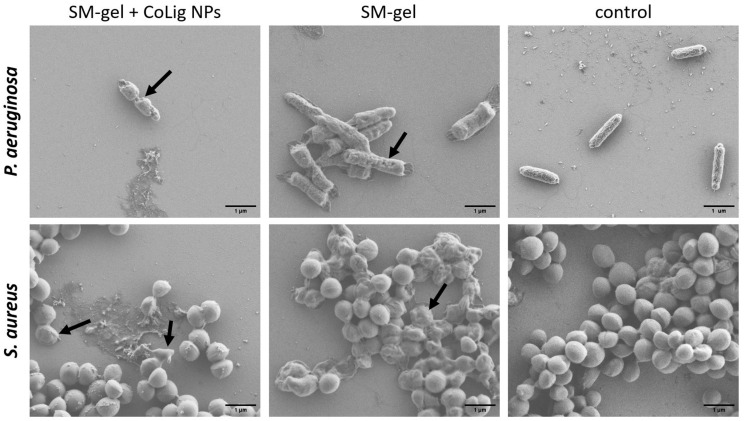
SEM images of bacteria after treatment with SM-gel with and without CoLig NPs. Arrows indicate cells with a damaged membrane.

**Figure 14 pharmaceutics-15-01717-f014:**
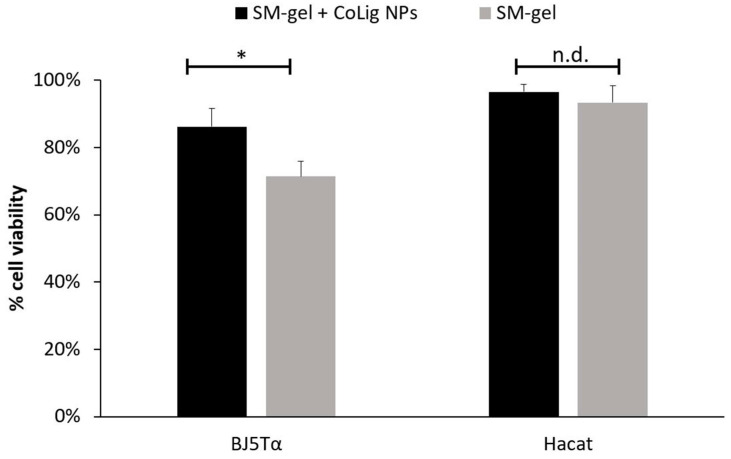
Cytotoxicity of SM-gel with and without CoLig NPs against human fibroblast (BJ-5ta) and keratinocytes (HaCaT). The * refers to *p* < 0.05, and n.d. refers to non-significative differences.

**Table 1 pharmaceutics-15-01717-t001:** CoLig NP characteristics: size obtained using DLS, NTA, or SEM analysis, and PDI and ζ potential obtained using DLS.

Technique	Size	PDI	ζ Potential
DLS	192 ± 1 nm	0.22 ± 0.01	−23.1 ± 0.8 mV
NTA	107 ± 38 nm	-	-
SEM	47 ± 7 nm	-	-

**Table 2 pharmaceutics-15-01717-t002:** MIC values in mg/mL for CoLig NPs and lignin NPs without cobalt (Lig-TA-LTH NPs) assessed using the broth dilution method.

	*S. aureus*	*P. aeruginosa*
CoLig NPs	1.2	2.4
Lig NPs	3.3	3.3

**Table 3 pharmaceutics-15-01717-t003:** Summary of rheological parameters from amplitude sweep test and self-healing strain test. Amplitude sweep test main parameters: G′ LVE (storage modulus within LVE), G″ LVE (loss modulus within LVE), and T_f_ (flow point, i.e., strain at which G′ and G″ curves cross). Self-healing strain tests parameter: percentage of G′ recovery (G′ recovered (%)) after 3 rupture cycles at 100% strain. Recovery was evaluated with respect to starting G′ values.

	G′ LVE (Pa)	G″ LVE (Pa)	T_f_ (%)	G′ Recovered (%) after 3 Cyclic Ruptures
SM-gel	4551.2	400.8	21.4	89%
SM-gel + CoLig NPs	7114.1	861.9	13.6	87%

## Data Availability

The data presented in this study are available on request from the corresponding author.

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
