# Peer review of "Lignin–Cobalt Nano-Enabled Poly(pseudo)rotaxane Supramolecular Hydrogel for Treating Chronic Wounds"

_pharmaceutics, 2023, doi:10.3390/pharmaceutics15061717_

Round 1

Reviewer 1 Report

This work presents a SM hydrogel-based drug delivery system that can be easily applied to the wound bed and delivers anti-inflammatory and antibacterial CoLig NPs. The authors demonstrated through experiments that the hydrogel has good antibacterial properties, anti-inflammatory capacity, injectability, self-healing properties, and linear release of the loaded cargo. Generally, the results could be valuable for treating chronic wounds. Some comments were suggested as follows.

1. Please consider using appropriate linking words to improve the coherence of the text. Additionally, it is recommended that the authors revise and polish the entire article to correct grammar mistakes.

2. In Figure 2, CoLig NPs morphology through SEM imaging, the shape of CoLig NPs is not very nice. Specifically, it is not a regular round nanoparticle.

3. It can be optimized by adjusting its color scheme to better display data and visual effects. In addition, a brief explanation of the figure can be provided in the annotation.

4. Mechanism descriptions for diabetic wound healing can be strengthened by citing 10.3390/ijms24032447; 10.1021/acsami.1c25014 and what are the advantages of the current work compared to published articles?

5. The "swelling" refers to the change in volume and is the ratio of the new volume to the old volume (in mL/mL). The increase in mass divided by the original mass is the "uptake" (in g/g). Unfortunately, the misuse of this terminology is very common.

No

Author Response

We would like to thank the reviewers for their comments and suggestions. Below is our point-by-point response.

Reviewer 1

Comments and Suggestions for Authors

This work presents a SM hydrogel-based drug delivery system that can be easily applied to the wound bed and delivers anti-inflammatory and antibacterial CoLig NPs. The authors demonstrated through experiments that the hydrogel has good antibacterial properties, anti-inflammatory capacity, injectability, self-healing properties, and linear release of the loaded cargo. Generally, the results could be valuable for treating chronic wounds. Some comments were suggested as follows.

  1. Please consider using appropriate linking words to improve the coherence of the text. Additionally, it is recommended that the authors revise and polish the entire article to correct grammar mistakes.

The text was revised and corrected.

  1. In Figure 2, CoLig NPs morphology through SEM imaging, the shape of CoLig NPs is not very nice. Specifically, it is not a regular round nanoparticle.

We modified the text as follows: “Finally, SEM images revealed an unregular round shape morphology of CoLig NPs, which is probably the result of the formation of compact clusters after the interaction between Co and Lig-TA-LTH (Figure 2).”

  1. It can be optimized by adjusting its color scheme to better display data and visual effects. In addition, a brief explanation of the figure can be provided in the annotation.

Figure 2 contrast was adjusted, and the caption was improved as follows: “CoLig NPs morphology assessed through SEM imaging. The CoLig NPs have an unregular round shape that is the result of clusters formation after Co and Lig-TA-LTH interaction.”

  1. Mechanism descriptions for diabetic wound healing can be strengthened by citing 10.3390/ijms24032447; 10.1021/acsami.1c25014 and what are the advantages of the current work compared to published articles?

The introduction was extended to underline the advantages of the proposed system and the additional literature has been cited.

  1. The "swelling" refers to the change in volume and is the ratio of the new volume to the old volume (in mL/mL). The increase in mass divided by the original mass is the "uptake" (in g/g). Unfortunately, the misuse of this terminology is very common.

The word “swelling” was substituted with “uptake”.

Reviewer 2 Report

After reading this manuscript from the beginning to the end, I found the manuscript lacking enough novelty to merit publication in Pharmaceuticals. Supramolecular hydrogels (SM) have been reported in the authors' previous works (J. Mater. Chem. B 797 2020, 8, 7696–7712; Materials Science and Engineering: C 2021, 127, 112194), and cobalt/lignin nanoparticles (CoLig NPs) were obtained according to the method proposed by Morenaet al (Ind. Eng. Chem. Res. 2020, 59, 4504–4514). If these CoLig NPs/SM nanocomposites were first reported here and applied to bacterial inhibition and cytotoxicity, it will be novelty. Meanwhile, the researches combined with or without metal NPs on manipulating antibacterial properties and cytotoxicity have been widely carried out. The explorations on chronic wounds in this study are too preliminary. Overall, I think this manuscript can be reconsidered after major revision.

Some points would be suggested to be improved before authors' future resubmission.

1. The explorations on bacterial inhibition and cytotoxicity in this study are too preliminary. It is suggested that research at the molecular level can be added.

2. Information on the apparent photography and topological morphology of the prepared CoLig NPs/SM composites should be provided and discussed in the main text.

3. The authors should include the data and explanations for all the weight ratios of CoLig NPs/SM composites or reason for explaining physical properties with specific ratio should be included.

4. X-ray photoelectron spectroscopy (XPS) and transmission electron microscopy (TEM) measurements of these composites are important for interfacial bonding characteristics and microstructural determination of these CoLig NPs/SM composites. If possible, these results should be involved in the main text.

5. In order to confirm the results of cytotoxicity test for CoLig NPs/SM composites on BJ-5ta and HaCaTcells, live/dead discrimination of cancer cells by flow cytometry was recommended to verify that point.

6. I am highly interested in the antibacterial activity of these composites. However, the antibacterial mechanism of the resulting composites still remains unclear.

7. In addition, influence of concentration of CoLig NPs/SM composites on the viability of S. aureus and P. aeruginosa should be discussed in the main text.

8. Do not refer to any in vitro study as "chronic wounds". Chronic wounds is an in vivo process that cannot be recapitulated in vivo. You may present as In Vitro Antibacterial and Cell Migration Assays or as appropriate.

Author Response

We would like to thank the reviewers for their comments and suggestions. Below is our point-by-point response.

Reviewer 2

Comments and Suggestions for Authors

After reading this manuscript from the beginning to the end, I found the manuscript lacking enough novelty to merit publication in Pharmaceuticals. Supramolecular hydrogels (SM) have been reported in the authors' previous works (J. Mater. Chem. B 797 2020, 8, 7696–7712; Materials Science and Engineering: C 2021, 127, 112194), and cobalt/lignin nanoparticles (CoLig NPs) were obtained according to the method proposed by Morenaet al (Ind. Eng. Chem. Res. 2020, 59, 4504–4514). If these CoLig NPs/SM nanocomposites were first reported here and applied to bacterial inhibition and cytotoxicity, it will be novelty. Meanwhile, the researches combined with or without metal NPs on manipulating antibacterial properties and cytotoxicity have been widely carried out. The explorations on chronic wounds in this study are too preliminary. Overall, I think this manuscript can be reconsidered after major revision.

Although we did not provide a full biological characterization of the complete system, the manuscript describes a novel application in which we merge a SM-hydrogel with CoLig NPs. To this aim, the SM-gel (the chemistry of which was previously published) was optimized in terms of polymer content to better fit the application in chronic wound. In particular, the uptake capacity of the system was studied anew and the SM-gel chosen parameters were the one granting better durability and mechanical properties. Moreover, although CoLig NPs were obtained by modifying a pre-existing protocol, the addition of LTH in the lignin-based material and the choice of cobalt as a metal still represent a novelty, considering that cobalt is poorly investigated compared to other metals in terms of antibacterial activity and applicability in the biomedical field. Therefore, we believe that the content of the manuscript is enough innovative reporting a new combination of materials. Nevertheless, additional experiments were added to strengthen the biological characterization.

Some points would be suggested to be improved before authors' future resubmission.

  1. The explorations on bacterial inhibition and cytotoxicity in this study are too preliminary. It is suggested that research at the molecular level can be added.

Indeed, the biological characterizations of the system in the paper are preliminary. However, we pretended to give more weight to the characterization of the supramolecular system given that the special issue is focused on this aspect.

  1. Information on the apparent photography and topological morphology of the prepared CoLig NPs/SM composites should be provided and discussed in the main text.

The result of the CoLig NPs integration is a homogeneous hydrogel with brown color (related to the presence of lignin). To better clarify this point, an image was added to the supplementary material (Figure S4). Moreover, the interaction between the polyurethane chains and CoLig NPs was studied through FTIR-ATR spectroscopy and the recorded spectra were added to the discussion (Figure 11).

  1. The authors should include the data and explanations for all the weight ratios of CoLig NPs/SM composites or reason for explaining physical properties with specific ratio should be included.

The parameters to obtain the hydrogel were optimized to be 3 % w/v of polymer content with a ratio CHP407:SHF68 of 80:20, and 10 % w/v of alpha-cyclodextrins. These parameters were selected to be optimal in terms of hydrogel stability and mechanical properties. The concentration of NPs inside the gel was selected to match the antibacterial activity for both the bacteria strains tested. The main text was improved to clarify this point.

  1. X-ray photoelectron spectroscopy (XPS) and transmission electron microscopy (TEM) measurements of these composites are important for interfacial bonding characteristics and microstructural determination of these CoLig NPs/SM composites. If possible, these results should be involved in the main text.

Unfortunately, we were not able to perform TEM due to current unavailability of the microscope. However, TEM images of different lignin-based NPs are reported in previous works (https://pubs.acs.org/doi/10.1021/acs.iecr.9b06362 and https://doi.org/10.1021/acsami.0c22301). We added these citations to the text to support our observations on NPs morphology after SEM analysis. Due to the small dimension of the NPs it was not possible to observe NPs inside the hydrogel and analyse the NPs-gel interaction through SEM. However, the completely different structure observed in the SM-gel loaded with the particles clearly suggested the occurrence of interactions among the system constituents, in accordance with rheological test results (Figure 9) and the ATR-FTIR spectra added to the revised version of the manuscript (Figure 11). With regard to XPS spectroscopy, the instrument was not available in our institute and it probably represents a quite difficult technique to implement due to the porous nature of the samples under investigation.

  1. In order to confirm the results of the cytotoxicity test for CoLig NPs/SM composites on BJ-5ta and HaCaT cells, live/dead discrimination of cancer cells by flow cytometry was recommended to verify that point.

Cytotoxicity of the hydrogel with and without NPs was performed and added to the paper showing the cytocompatibility of the system.

  1. I am highly interested in the antibacterial activity of these composites. However, the antibacterial mechanism of the resulting composites still remains unclear.

SEM imaging of the bacterial strains treated with the SM-gel loaded with NPs was performed proving an interaction with the bacterial membrane.

  1. In addition, influence of concentration of CoLig NPs/SM composites on the viability of S. aureus and P. aeruginosa should be discussed in the main text.

Instead of evaluating the influence of the concentration of CoLig NPs/SM composites on bacterial viability, the antibacterial activity was confirmed by using the 24h-release from the hydrogel. This was performed in order to match the antibacterial result with the result of the NPs release rate from the hydrogel at 24h. Furthermore, the NPs showed concentration-dependent toxicity toward bacteria, so it is expected that a higher amount of CoLig NPs in the SM composites would present a higher antibacterial effect.

  1. Do not refer to any in vitro study as "chronic wounds". Chronic wounds is an in vivo process that cannot be recapitulated in vivo. You may present as In Vitro Antibacterial and Cell Migration Assays or as appropriate.

The text was modified as suggested.

Reviewer 3 Report

1.      Although this article is interesting, none of the characterizations of the material are actually in qualitative levels. Any qualitative results (like the morphology of the treated cell, the morphology of the treated Gram-negative and Gram-positive strains) will strongly support the functional novelty of their material. For guidance, the following paper can be cited and discussed when doing the revision: https://doi.org/10.1016/j.colsurfb.2022.113096; https://doi.org/10.1016/j.ijbiomac.2022.11.184.

2.      Also, this work has no statistical analysis description. The means ± standard deviation was employed to express experimental data. And one-way analysis of variance was employed to evaluate differences. (*p < 0.05, **p < 0.01, ***p < 0.001).

3.      The conclusion section has four paragraphs, which should be revised.

Author Response

We would like to thank the reviewers for their comments and suggestions. Below is our point-by-point response.

Reviewer 3

Comments and Suggestions for Authors

  1. Although this article is interesting, none of the characterizations of the material are actually in qualitative levels. Any qualitative results (like the morphology of the treated cell, the morphology of the treated Gram-negative and Gram-positive strains) will strongly support the functional novelty of their material. For guidance, the following paper can be cited and discussed when doing the revision: https://doi.org/10.1016/j.colsurfb.2022.113096; https://doi.org/10.1016/j.ijbiomac.2022.11.184.

Images of the cells treated with CoLig NPs were added to the supplementary material (Figure S1). The images confirmed the results of the viability assay. Moreover, the morphology of bacteria treated with CoLig NPs was observed through SEM imaging and added to the discussion. The suggested papers were added in the discussion section.

  1. 2. Also, this work has no statistical analysis description. The means ± standard deviation was employed to express experimental data. And one-way analysis of variance was employed to evaluate differences. (d. not significative difference, *p < 0.05, **p < 0.01, ***p < 0.001).

The statistical analysis is described in Material and methods as follows “Results are reported as mean ± standard deviation. Statistical analysis was performed through JASP for Windows 10 (version 0.9.2). To compare results, one-way ANOVA analysis followed by Bonferroni’s multiple comparison test was performed. The statistical significance was assessed as described: n.d. not significative difference, * p < 0.05, ** p < 0.01, *** p < 0.001.”

  1. The conclusion section has four paragraphs, which should be revised.

The conclusion section was revised and the length reduced.

Round 2

Reviewer 2 Report

Through the revision, most of my comments were addressed, and the quality of manuscript significantly improved with additional experiments. However, one issue regarding the antibacterial mechanism of the resulting composites has not been explicitly discussed. Therefore, this manuscript is required minor revision for the publication in Pharmaceutics.

Author Response

The part describing the antibacterial mechanism of the composite was expanded and better discussed.

Reviewer 3 Report

No comments.

Author Response

Thank you for reviewing the paper again